# Zebrafish: A Model Deciphering the Impact of Flavonoids on Neurodegenerative Disorders

**DOI:** 10.3390/cells12020252

**Published:** 2023-01-07

**Authors:** Kamel Mhalhel, Mirea Sicari, Lidia Pansera, Jincan Chen, Maria Levanti, Nicolas Diotel, Sepand Rastegar, Antonino Germanà, Giuseppe Montalbano

**Affiliations:** 1Zebrafish Neuromorphology Lab., Department of Veterinary Sciences, University of Messina, Via Giovanni Palatucci snc, 98168 Messina, Italy; 2Institute of Biological and Chemical Systems-Biological Information Processing (IBCS-BIP), Karlsruhe Institute of Technology (KIT), Campus North, Hermann-von-Helmholtz-Platz 1, 76344 Eggenstein-Leopoldshafen, Germany; 3Université de la Réunion, UMR 1188 Diabète Athérothrombose Thérapies Réunion Océan Indien (DéTROI), Plateforme CYROI, F-97490 Sainte-Clotilde, France

**Keywords:** zebrafish, flavonoids, neurogenesis, neuroplasticity, brain, Alzheimer, neurodegeneration, neuroinflammation, antioxidant, neuropathology

## Abstract

Over the past century, advances in biotechnology, biochemistry, and pharmacognosy have spotlighted flavonoids, polyphenolic secondary metabolites that have the ability to modulate many pathways involved in various biological mechanisms, including those involved in neuronal plasticity, learning, and memory. Moreover, flavonoids are known to impact the biological processes involved in developing neurodegenerative diseases, namely oxidative stress, neuroinflammation, and mitochondrial dysfunction. Thus, several flavonoids could be used as adjuvants to prevent and counteract neurodegenerative disorders such as Alzheimer’s and Parkinson’s diseases. Zebrafish is an interesting model organism that can offer new opportunities to study the beneficial effects of flavonoids on neurodegenerative diseases. Indeed, the high genome homology of 70% to humans, the brain organization largely similar to the human brain as well as the similar neuroanatomical and neurochemical processes, and the high neurogenic activity maintained in the adult brain makes zebrafish a valuable model for the study of human neurodegenerative diseases and deciphering the impact of flavonoids on those disorders.

## 1. Introduction

Over the past century, advances in biotechnology, biochemistry, and pharmacognosy have renewed interest in natural medicines, which have many advantages over synthetic drugs [1], namely, a strong binding affinity, high efficacy, lower toxicity, and fewer side effects, and are therefore safer and have few drawbacks [2,3]. This interest is based on the growing awareness that a large number of secondary metabolites, namely polyphenols [4], alkaloids [5], and terpenoids [6], play an important role in human health.

Flavonoids are a substantial family of polyphenolic secondary metabolites that have attracted great interest in scientific research in the recent decade. They are able to regulate the expression of multiple genes and modulate many molecular pathways involved in various biological mechanisms. For example, they exert anti-inflammatory activities by reducing the formation of reactive oxygen species (ROS) and decreasing the expression of some inflammatory mediators [7,8]. Thus, they prevent pathological processes such as aging [9], cancer [10], cardiovascular diseases [11], inflammation-related diseases [12], and neurodegenerative diseases [13].

At the neurological level, there is growing evidence of the link between a flavonoid-rich diet and improved cognitive abilities [14,15], probably because flavonoids influence signaling pathways involved in neuronal plasticity, learning, and memory (i.e., cAMP response element-binding protein/extracellular signal-regulated kinase/brain-derived neurotrophic factor (CREB/ERK/BDNF) or serine/threonine-specific protein kinase/protein kinase B (Akt/PKB) [16,17]. Furthermore, it has been shown that several flavonoids could be used as adjuvants in the prevention and cure of neurodegenerative disorders such as Alzheimer’s (AD) [18,19] and Parkinson’s [20]. 

Although neurogenesis in adulthood was considered impossible in mammals for almost a century, new data have clearly demonstrated that the brains of adult vertebrates, including humans, maintain a significant neurogenic activity in discrete regions throughout life [21,22,23,24]. Indeed, after embryonic development, neurogenesis is mainly restricted to the subgranular zone (SGZ) in the hippocampus and to the subventricular zone (SVZ) in the forebrain [25,26,27,28]. In these neurogenic niches, the alterations in the production of new neurons or the loss of homeostasis between new neuron genesis and apoptosis have been linked to psychiatric and neurodegenerative disorders [29,30]. Given that flavonoids inhibit pro-apoptotic mechanisms and enhance neuronal plasticity [31,32], they have attracted great interest in combating neurodegenerative diseases. These phytochemical compounds have been shown to reduce neuroinflammation [33] via the modulation of inflammatory cytokines released by mixed glial cells (astrocytes and microglia) [34]. They also modulate the activity of several enzymes involved in oxidative stress, such as superoxide dismutase (SOD), cyclooxygenase-1 (COX-1), and cyclooxygenase-2 (COX-2) [35]. In addition, numerous studies have shown that some flavonoids can interfere with the formation and accumulation of neurotoxic proteins, such as amyloid β-protein 42 (Aβ42) [36] and α-synuclein [37], which are responsible for the progression of neurodegenerative diseases.

Zebrafish (*Danio rerio*) is an interesting model organism that can offer new opportunities in the research of beneficial effects of flavonoids, thanks to a combination of interesting aspects and several experimental advantages. Firstly, zebrafish are small, robust fish that are easy to maintain and inexpensive to grow, allowing the screening of large libraries of natural products and drugs [38,39]. Secondly, the optically transparent zebrafish are suitable for optical monitoring and manipulation from the time of external fertilization until organogenesis. Thirdly, the high genome homology of 70% to humans makes zebrafish a valuable model for the study of human diseases [39,40]. Finally, the zebrafish is a valuable model for the study of neurodegenerative diseases, as the organization of the zebrafish brain and the human brain is largely similar, and the neuroanatomical and neurochemical processes are very similar [41]. The use of zebrafish in the research of new treatments based on natural compounds could accelerate the discovery of pharmacological adjuvants without side effects on human health.

This review aims to decipher the potential of flavonoids for treating neurodegenerative diseases and describe the neurogenic and regenerative features of the zebrafish brain compared to other vertebrates, especially the human brain. We first reported the different subclass of flavonoids, their bioavailability in the brain, and their effects on neurodegenerative disorders. In the second part, we compare the neuroanatomy and neurochemistry of zebrafish nervous system (NS) of relevance to other vertebrates’ NS and the effects of flavonoids on its neurodegenerative disorders.

## 2. Flavonoids

Flavonoids are a large family of polyphenolic secondary metabolites of plants that are synthesized from phenylalanine via the phenylpropanoid pathway. They are a series of pigments that impart different colors to fruits and vegetables and are found in green plants, tea, coffee, and wine [42]. Therefore, flavonoids are an important component of the human diet [43]. These natural compounds have attracted increasing interest because they possess various biochemical properties useful for maintaining good health. They can strongly absorb UV radiation, and their accumulation in the leaf epidermis suggests a specific function to protect against the damage that this radiation causes to cell DNA [44,45]. 

### 2.1. Chemical Structure and Subclasses of Flavonoids

The plant enzyme chalcone synthase (CHS) is responsible for chalcone biosynthesis, the basic structure of all flavonoids (Figure 1). The general structure of flavonoids consists of a diphenyl propane skeleton composed of two benzene rings (A and B), connected by a chain of three carbon atoms forming a pyran ring I (oxygen-containing heterocyclic ring) [40]. Indeed, in the presence of one molecule of p-Coumaroyl CoA and three of malonyl-CoA, the CHS catalyzes a series of sequential decarboxylations and condensations, during which a polyketide intermediate is formed that undergoes cyclizations and aromatization, leading to the formation of the A ring, and the resulting chalcone structure. The product of the above reactions is naringenin chalcone (2′,4,4′,6′-tetrahydroxychalcone, C_15_H_12_O_5_), 6′-hydroxychcone, and the first flavonoid produced [46]. Naringenin chalcone, (2S)-naringenin, and its derivative dihydro-kaempferol (dihydroflavonol) are central intermediates in flavonoid biosynthesis, being branching points from which the synthesis of distinct flavonoid subclasses departs. Through the action of several enzymes such as isomerase, reductase, hydroxylase, glycosyltransferase, and acyltransferase that modify the structure of the C ring and the degree of unsaturation and oxidation, all varieties of chemical subclasses of flavonoids are formed: isoflavones, flavones, flavonols, flavanones, flavanols, and anthocyanidins (Figure 1) [47].

#### 2.1.1. Isoflavones

Isoflavones are an important subclass of flavonoids that consist of a 3-phenylchrome backbone derived from the 2-phenylchrome backbone via an aryl migration mechanism [48]. In contrast to most flavonoids, where the B ring binds to the C ring in position 2 in isoflavones, the B ring binds to the C ring in position 3 in isoflavones (Figure 1). Isoflavones are almost found in plants belonging to the family of Leguminosae or Fabaceae, in particular in soybean [49], and are involved in plant-microbe interactions and defense mechanisms [50]. Isoflavones are structurally similar to estrogens, particularly to 17β-estradiol. This similarity confers them the ability to modulate estrogen signaling through the binding of nuclear estrogen receptors-α and β (ERs). They are consequently considered “phytoestrogens” due to their hormone-like mechanism [51]. The most abundant isoflavones are genistein (4′,5,7-trihydroxyisoflavone), and daidzein.

Additionally, isoflavones play a central role in regulating mechanisms of adipogenesis and osteogenesis; in particular, genistein shows modest selectivity for ERβ [52]. It has different dose-dependent biological effects. At low concentrations, genistein acts as an estrogen (positive effect on osteogenesis and negative effect on adipogenesis), while at high concentrations (>1 μM), it acts as a ligand of PPARγ by stimulating the transcription of its target genes; it up-regulates adipogenesis [53]. They are also characterized by high antioxidant power, probably mediated by moderating the nuclear factor erythroid 2-related factor 2 (Nrf2)-ARE pathway, a mechanism that stimulates the antioxidant defense system and phase 2 detoxifying enzymes as NAD(P)H Quinone oxidoreductase 1 (NQO1), and Heme oxygenase 1 (HO-1) [54,55].

#### 2.1.2. Flavonols

The basic chemical structure of flavonols is the 3-hydroxyflavone, characterized by a double bond between C2=C3 and the presence of a ketone group in C4. In addition, the 3-hydroxyl group can be attached to a sugar (glycosylated), usually glucose or rhamnose [56]. Flavonols are found abundantly in fruits and vegetables such as capers, onion, broccoli, berries, and grapes but also in green and black tea. The most common flavonols are quercetin, kaempferol, myricetin and isorhamnetin. 

Flavonols have attracted great interest in the field of biomedical research for their capacity to improve conditions of hypertension, inflammation, and cardiovascular risk [57]. Quercetin and its derivatives have demonstrated a significant protective effect on low-density lipoproteins (LDL) against oxidative modification [58]. They could also reduce the risk of neurodegenerative disorders such as AD [59] due to their antioxidant capacity, including the one of the o-dihydroxy (catechol) in the B ring. It acts as a radical target site for the double bond between C2=C3, cojoined to the 4-keto group, which allows the delocalization of electrons from the B-ring and the 3- and 5-hydroxyl groups, allowing the strong adsorption of radicals and the maximum radical scavenging power (Figure 1) [60].

#### 2.1.3. Flavones

Flavones are colorless-to-yellow flavonoids of the C_15_H_10_O_2_ chemical formula, derived *from flavanone* by the introduction *of* a *double bond between* C-2 and C-3 and a 4-keto group on the C ring. The majority of flavones present a hydroxyl group in position 5 of the A benzene ring and less often in position 7 of the A benzene ring or 3′ and 4′ of the B ring [61]. The most important flavones are luteolin and apigenin. 

Luteolin is mainly found in carrots, olive oil, peppers, and in medicinal plants such as thyme, rosemary, and oregano [62]. Luteolin has shown several biological effects, including the prevention and treatment of cancer through both pro-apoptotic and anti-angiogenic effects [63,64]. Additionally, luteolin has high antioxidant power and radical scavenging activity as the basis of its neuroprotective action [65]. 

Apigenin (Figure 1), however, is *present* in tea, chamomile, onions, oranges, and many other fruits and vegetables and is responsible for its anti-inflammatory, anti-bacterial, and anti-spasmodic action [66,67]. In recent years, interest in apigenin has grown, especially for its chemopreventive and anti-cancer properties and its capacities to modulate phosphoinositide-3-kinase (PI3K)/protein kinase B (Akt), mitogen-activated protein kinase (MAPK)/extracellular signal-regulated kinase (ERK), nuclear transcription factor-κB (NF-κB), and Wnt/β-catenin pathways, involved in cell proliferation and survival, as well as in the processes of migration and metastasis [68]. 

#### 2.1.4. Flavanones

Flavanones (dihydroflavones) have a skeleton of 2,3-dihydro in flavonoids C_6_–C_3_–C_6_ structure and, unlike flavones, lack the double bond between positions 2 and 3 (Figure 1). They can be multi-hydroxylated, and the hydroxyl groups, in turn, can be methylated and glycosylated [69]. They are found in many plant species, especially in Compositae, Leguminosae, and Rutaceae, and can be detected in all parts of the plant: roots, branches, bark, leaves, fruits, seeds, and peel. The most important flavanones are eriodictyol, hesperetin and naringenin.

The antioxidant activity of flavanones depends on the number and spatial position of phenolic OH groups. Hesperetin and naringenin, the aglycones of the flavanone glycosides hesperidin and naringin, occur naturally in citrus species such as lemon, orange, lime, grapefruit, and bergamot. Hesperidin and its aglycone hesperetin have shown a great capacity to reduce oxidative stress and inflammation by regulating the nuclear factor erythroid 2- related factor 2 (Nrf2)/Toll-like receptor 4 (TLR4)/NF-κB signaling pathways [70].

Naringenin and its glycoside naringin as well exert a potent anti-oxidant effect, yet naringenin aglycone has shown greater antioxidant capacity than its glycoside [71]. Besides its antioxidant effects, naringenin effectively contrasts the inflammatory processes by reducing the expression of pro-inflammatory cytokines such as interleukin-1β(IL-1β), IL-6, IL-8, and tumor necrosis factor-alpha (TNF-α) [72], and targeting cell death in tumors, via increased expression of caspase-3 and subsequent activation of the caspase cascade, and inhibiting cell migration by reducing matrix metalloproteinase-2 (MMP-2) and MMP-9 expression and *regulating cell migration* [73].

#### 2.1.5. Flavanols 

Flavanols (flavan-3-ol), often called catechins, are characterized by the absence of the carbonyl group at position 4, and a saturated and disubstituted at positions 2 and 3 pyran ring [69]. They are found in foods and beverages of plant origin, such as cocoa, berries, apples, red grapes, and tea, usually accumulated in seeds or in the peel; for this reason, the diet intake of flavanols is limited. 

Among flavanols, catechin, a (*2R*,*3S*)-2-(3,4-dihydroxyphenyl)-3,4-dihydro-2*H*-chromene-3,5,7-triol with two steric forms of (+)-catechin and its enantiomer, is one of the most common. Several studies show the effects of cocoa flavanols in the reduction of the risk of developing chronic pathologies, including cardiovascular diseases [74,75], metabolic diseases [76], and cancer [77]. The preventive and therapeutic effects of catechins against chronic diseases are exerted via their antioxidant properties. These antioxidant capacities are related to the redox properties of their phenolic hydroxyl groups [78] or to the increase in the activity of key enzymes in the scavenging of ROS, such as catalase (CAT), SOD, and glutathione peroxidase (GSH) (Figure 1) [79].

#### 2.1.6. Anthocyanidins

Anthocyanidins are a class of water-soluble flavonoids characterized by a specific absorption in the visible range. They are naturally present as glycosides called structural derivatives of the flavylium cation (2-phenylbenzopyrilium ion), containing a C-15 backbone structure arranged in two C-6 benzyl rings (A and B) and a heterocyclic ring (C) [80]. They provide pH-dependent colors to the plant parts, such as red, blue, and purple [81]. Anthocyanidins are found abundantly in red fruits such as berries (blackcurrants, blueberries, and strawberries), vegetables, tea, nuts, olive oil, cocoa, and cereals. The most common anthocyanidins are cyanidin, pelargonidin, delphinidin, malvidin, petunidin, and peonidin. They are currently studied for their antioxidant [82], anti-angiogenic action [83], and the capacity to improve neuronal and cognitive brain function [84]. 

### 2.2. Bioavailability of Flavonoid in the Brain

The free exchange between blood and interstitial fluid occurs in nearly all organs of the body except the brain capillaries. The cerebral endothelial cells exhibit tight junctions (i.e., claudin 5, zona occludens 1, and occludins) and form with pericytes and astrocytes the brain blood-barrier (BBB), a physiological barrier restricting the movement of molecules and cells between the blood and the brain. This important interface has a protective function, preventing toxic and infectious substances from entering the brain [85,86]. Flavonoids have been detected in the blood after consumption of foods rich in these components, suggesting their bioavailability also in the periphery of tissues. However, this evidence is not sufficient to prove their presence in the brain and CNS and, consequently, their ability to overcome the BBB [85]. 

Bioavailability studies using orally supplemented flavonoids or flavonoids rich diet demonstrated their presence in various brain tissues such as the hypothalamus, superior colliculus, cerebellum, and striatum, and in limbic system structures such as the cortex and hippocampus [87,88]. Both the cortex and hippocampus are important for memory formation and adversely affected by aging and neurodegenerative diseases [89]. Recent studies in the CNS system indicated the presence of flavonol (e.g., (−)-epicatechin), flavanones (e.g., hesperetin), and flavone (e.g., baicalein) and their metabolites in the CNS system following their oral administration [88]. These encouraging data suggest that we can advance the study of flavonoid bioavailability into a new area where CNS disorders may be the real target of their biological activity in vivo. 

Many studies have explored the kinetics and extent of flavonoid absorption by measuring plasma concentrations after the ingestion of a single dose of flavonoids provided as whole food/beverage, plant extract, or pure compound [90,91,92,93,94,95,96,97,98,99]. We have summarized the different bioavailability measures, including the maximal plasma concentration (C_max_), time to reach C_max_, and the elimination half-life reported in the above-mentioned studies (Table 1). The results show wide variability in the bioavailability of the different flavonoids.

Isoflavones, followed by catechins, flavanones, and quercetin glucosides, are the most well-absorbed polyphenols, with different kinetics depending on the site of intestinal absorption. The least well-absorbed flavonoids are the anthocyanins. These data could be useful for the design and interpretation of studies investigating the health effects of flavonoids [99]. Quercetin glucosides, catechins, and anthocyanins, which are absorbed in the stomach or the small intestine, reached C_max_ at ~1.5–2 h, while rutin and the flavanones hesperidin and naringin, which are absorbed after the release of the aglycones by the microflora, reached C_max_ at ~5.5 h [99]. 

## 3. The Roles of Flavonoids in Neurodegenerative Diseases

Neurodegenerative diseases affect nerve cells and NS, as well as non-neuronal cells, impacting motor, sensory, and/or cognitive functions [30,100]. Currently, the specific causes underlying neurodegenerative disease are not well understood. Several cellular and molecular mechanisms are thought to underlie age-related neurodegeneration, including oxidative stress, mitochondrial dysfunction, deposition of neurotoxic protein aggregates, and chronic neuroinflammation leading to excitotoxicity and neuronal apoptosis [101,102]. Given the pleiotropic effects of flavonoids, they could be considered interesting molecules that scavenge free radicals, modulate the brain’s immune system, inhibit neuroinflammation, enhance neuroplasticity and promote neuronal survival and differentiation (Table 2). The molecular mechanisms underlying the neuroprotective and neurogenic effects of this polyphenolic subclass are described in detail below.

### 3.1. Neuroinflammation in Neurodegenerative Disorders

Inflammation appears to be one of the fundamental mechanisms involved in the progression of neurodegenerative diseases [117]. Although inflammation is not a direct trigger of neurodegenerative diseases, it is not surprising that persistent inflammation influences their progression through several mechanisms, such as initiating inflammatory responses that contribute to neuronal dysfunction and death [117] and proteostasis disturbances through the induction of ROS or reactive nitrogen species (RNS), leading to protein misfolding [118]. Furthermore, activated microglia release pro-inflammatory cytokines such as TNFα and IL-1β, which can have direct excitotoxic effects on synapses inducing synapse loss [119,120]. Uncontrolled activation of astrocytes (versatil CNS cells) and microglia (the resident macrophages of the brain parenchyma) under brain injury induces their active participation in self-enduring neuronal damage cycles (Figure 2) [121,122,123,124,125]. Several critical mechanisms through which damaged neurons activate microglia were reported. In diseased brain, resting microglial cells proliferate quickly and persistently increase the expression of a significant number of marker molecules, including CD11b, CD68, and major histocompatibility complex I and II (MHC-I and II) molecules, and may be transformed into M1 or M2 macrophages [122]. Indeed, neurons are able to control microglial activation. Both direct excitatory signals and/or loss of inhibitory signals by neurons incite activation of microglia under pathological conditions and contribute to the inflammatory milieu of neurodegenerative disease. The released excitatory signals, listed as chemokines (CX3CL1, CCL21, and CXCL10), glutamate, purines (ATP and UTP), or MMP-3, control various aspects of microglia function [122]. Proteases, such as MMP-3, released following neuronal damage, are known to degrade the extracellular matrix (ECM) components and activate microglia to further propagate neuronal cell death. Besides MMP-3, damaged dopaminergic neurons release neuromelanin-activating microglia in the substantia nigra. Neuromelanin can be neurotoxic, inhibiting the function of dopaminergic neurons and proteasomes and inducing the production of toxic factors, namely TNF-α, IL-6, and nitric oxide (NO) (Figure 2) [122,126]. Astrocytes detect neuron-derived α-synuclein released by neurons and get activated. Such reactive astrocytes, as well as microglia, express proinflammatory cytokines (IL-1α, IL-1β, IL-6, IL-18, and colony-stimulating factors-1, 2, and 3), resulting in a strong inflammatory response beside changes in chemokine expression, such as CXC-type (CXCL-1, 2, 5, 10, 11, 12, 16), CC-type (CCL-3, 4, 5, 12, 20), and CX3C-type (CX3CL1) chemokines, which in turn incite glutamate release and the synthesis of cytokines and chemokines restart in astrocytes (Figure 2) [127,128]. 

Activated CD4^+^ T cells release multiple inflammatory factors, such as the Fas ligand. This aforementioned binds with the Fas receptor of astrocytes, causing the release of monocyte chemoattractant protein-1, one of the key chemokines, and various cytokines, including IL-6 and IL-8 [122,129]. Numerous scientific studies have shown that a chronic inflammatory state promotes the evolution of neurodegenerative diseases, such as Parkinson’s disease (PD), AD, Multiple Sclerosis, and Amyotrophic lateral sclerosis (ALS) [130,131,132,133]. In PD, for example, it has been postulated that the course of the disease worsens in case of excessive activation of microglia, overproduction of cytokines and other inflammatory mediators, as well as ROS (Figure 2) [134]. Further support for this theory comes from studies of in vivo imaging of microglial activation with the peripheral benzodiazepine receptor binding I and [^11^C]-(R)-PK11195 in positron emission tomography (PET) scan [135]. 

ROS can cause lipid peroxidation and protein oxidative modification at the cellular level, leading to the generation of 4-hydroxy-2-nonenal (4-HNE) [136]. Owing to its high reactivity, 4-HNE forms protein adducts that cause protein misfolding and disturbances in their function [118]. Furthermore, 4-HNE can induce carbonyl stress and reduce the cell’s antioxidant capacity. In AD, PD, ALS, and Huntington’s disease (HD), lipid peroxidation has been documented, and elevated levels of 4-HNE have been reported in AD patients [137]. Additionally, 4-HNE affects the enzymes implicated in the elimination of amyloid β-protein (Aβ), which are key enzymes of energy metabolism, including aldolase, enolase, aconitase, and ATP synthase [138], as well as enzymes involved in antioxidant defense, such as superoxide dismutase, heme oxygenase, and peroxiredoxins [118,139].

### 3.2. Neuroinflammation: Modulation by Flavonoids

Flavonoids could be useful to prevent and treat neuroinflammation by reducing microglial activation, modulating the mRNA and protein expression of inflammatory cytokines [140], inhibiting inducible nitric oxide synthase (iNOS) induction and subsequent NO production [141,142], as well as inhibiting reduced nicotinamide adenine dinucleotide phosphate (NADPH) oxidase in activated glia [143]. Additionally, flavonoids have been reported to reduce the expression of COX-2 [144] and increase the expression of BDNF, a key player of brain plasticity.

In a recent study performed in chicken brain, the flavonol quercetin inhibited the ROS/iNOS/NF-κB pathway, counteracting Cadmium treatment-induced neurotoxicity and thereby reducing necroptosis [145]. In addition, quercetin has been shown to significantly reduce the expression of inflammatory cytokines such as IL-6 and IL-1β [103]. 

It has also been reported extensively in the literature that estrogens play a role in protecting against inflammation and thus reduce the neuroinflammatory process [146] through the reduction of pro-inflammatory molecules and the regulation of microglial reactivity [147,148]. For this reason, particular focus was paid to the isoflavone genistein, a phytoestrogen, which has been shown to reduce neuroinflammation by decreasing the expression of Toll-Like 4 (TLR-4) and consequently inhibiting microglial polarization and promoting the phenotypic switch from M1 (phenotype associated with neurotoxicity) to M2 (phenotype that promotes the recovery of homeostasis) [115]. The polyphenolic flavone Luteolin has shown the capacity to inhibit microglia activation by reducing the release of inflammatory mediators and alleviate neuroinflammation induced in an in vivo brain hemorrhage model by downregulating TLR-4/TNF receptor-associated factor 6 (TRAF6)/NF-κB signaling [116].

### 3.3. Oxidative Stress in Neurodegenerative Diseases

The brain has a heavy demand for oxygen and thus consumes 20% more oxygen than other parts of the body. Additionally, it is well enriched in redox-active metals (copper and iron) that actively participate in ROS generation [149]. Although oxygen is involved in many cellular activities, coincidently, when present in excess, it has deleterious effects on neuronal cells via modulating the function of biomolecules, resulting from its univalent metabolic reduction status at the origin of ROS. Besides ROS (hydrogen peroxide (H_2_O_2_), superoxide anion (O_2_), and highly reactive hydroxyl radical (HO^•^), RNS such as NO are found to be actively involved in neurodegenerative disease (Figure 2) [149,150]. The RNS, such as NO, are also found to have a deleterious effect on neurons. Among the array of mediators released, increased ROS release and oxidative stress have been implicated in the induction and amplification of neurotoxicity and, consequently, in the pathogenesis of numerous neurodegenerative conditions [151,152]. Under circumstances of oxidative stress, the generation of the O_2_^−^ in the mitochondria is the first step in the formation and proliferation of other reactive oxygen species (Figure 2). These free radicals react with hydrogen peroxide through the iron-catalyzed Haber-Weiss reaction that generates the hydroxyl radical (OH^•^) [153]. ROS cause mitochondrial dysfunction and apoptosis through irreversible damage to the cellular macro-molecules that are associated with the alteration of mitochondrial membrane functions [154,155].

In addition, oxidative stress causes proteotoxicity by altering protein structure or affecting the nascent polypeptide chain folding (Figure 2) [156,157]. For example, exposure to pesticides induces aggregation of α-synuclein and amyloid beta, as well as an increase in tau hyperphosphorylation, thereby raising the risk of developing neurodegenerative diseases [158].

### 3.4. Antioxidant Effect of Flavonoids in Neurodegenerative Diseases

Considering the above-mentioned data, antioxidants would seem to be useful in preventing and counteracting neurodegenerative diseases. In recent years, numerous studies have pointed up the antioxidant properties of flavonoids due to their chemical characteristics. In fact, as mentioned earlier, flavonoids possess in their chemical structure numerous -OH groups that allow them to act as H or electron donors (low potential redox). For this reason, they act as free radicals scavengers and metal chelators [159]. In addition, flavonoids, such as hesperetin, affect the expression of antioxidant enzymes such as GSH and SOD by modulating the nuclear factor erythroid 2-related factor 2 (Nrf2)-ARE pathway [107].

A recent study compared the in vitro and in vivo antioxidant activity of six different flavonoids with similar structures, including quercetin, rutin, taxifolin, epicatechin, epigallocatechin, and procyanidin B2 [104]. The antioxidant activities of the six flavonoids were estimated in vitro by DPPH and ABTS^+^ free radical scavenging assays and compared with the antioxidant action of Vitamin C. The results demonstrate a high capacity of flavonoids to scavenge ROS in a dose-dependent manner, which was significantly greater than that of Vitamin C. Among the six flavonoids tested, procyanidin B2 showed the highest scavenging capacity in relation to the number of -OH groups present, thus confirming the correlation of antioxidant activity with chemical structure. To evaluate the antioxidant potentials of flavonoids in vivo, mice with premature aging induced by six weeks of intraperitoneal injection of D-galactose were used [160]. The activities of the antioxidant defense enzymes SOD, CAT, and glutathione peroxidase (GSH-Px) were significantly decreased in D-galactose-treated mice compared with control mice, while there is evidence of increased serum levels of NO and malondialdehyde (MDA). However, these changes were reversed differently by treatment with the flavonoids, revealing that the six flavonoids have different antioxidant potentials in D-galactose-treated mice, in the following order: procyanidin B2 > epicatechin, epigallocatechin > epicatechin, quercetin > taxifolin, quercetin > rutin [104]. Overall, the in vitro and in vivo findings are coherent and show a close correlation between the characteristic chemical structure of the individual flavonoid and its antioxidant capacity.

### 3.5. Neuroplasticity: A Combined Process of Neurogenesis and Synaptogenesis

Neuroplasticity is a combined physiological process taking place in the brain for its entire life to reestablish the functional and structural organization of neurovascular networks in response to intrinsic or extrinsic stimuli regulated by the qualities of the tissue environment. This process includes a synergistic effect between neurogenesis, synaptogenesis, and neurochemical changes in the CNS [161]. Structural plasticity, which refers to the diverse modifications in the anatomical properties of the neural tissue (the number, location, and size of spines, dendritic and axonal branching patterns), has been reported in the developing brain and/or as a result of several weeks or much shorter time scale learning [162,163,164]. The compensatory functional plasticity, however, occurring in a damaged brain, begins in critical conditions related to inflammation, degeneration of nerve fibers, apoptosis, edema, and metabolic disorders. Synaptogenesis is based on improving existing synaptic pathways and forming new connections resulting in restored functions of the damage [165]. Several brain diseases, such as neurological and psychiatric ones, are the consequences of the dysregulation of adult hippocampal neurogenesis [166,167,168]. 

The phospholipase C/inositol trisphosphate/Ca^2+^/calmodulin-dependent protein kinase II (PLC/IP3/CAMKII), MAPK/ERK, and PI3K/Akt pathways are among the most important signaling pathways involved in brain plasticity, which are activated by divers growth factors namely BDNF, nerve growth factor (NGF), insulin-like growth factor 1 (IGF1), fibroblast growth factor (FGF), and Wnt [161,169]. Neurotrophic factors bind to two classes of receptor tropomyosin receptor kinase (Trk) and the p75 receptor belonging to the tyrosine kinase receptors family and the tumor necrosis factor receptor superfamily, respectively. The p75 receptor exercises its functions mainly via interactions with other effector proteins [161,170]. Through these two receptor classes, neurotrophins regulate the neuron’s survival and ensure their proper development and normal function. The binding of growth factors to Trk receptors induces their dimerization and phosphorylation of their cytoplasmic domain, which in turn activates the cytoplasmic domain tyrosine kinases [161,171]. The activation of signaling pathways through Trk receptors involves adapter proteins containing Src Homology 2 (SH2) or phosphotyrosine-binding (PTB) domains, which induce intracellular signaling PI3K and PKB/Akt pathways. These pathways lead to the activation of the expression of genes implicated in brain plasticity, promoting cell survival through the activation of anti-apoptotic mechanisms [17,161,172]. The phosphorylation of the Trk receptor pilot, as well as the activation of PLC-1, which successively catalyzes the hydrolysis of phosphatidylinositol-4,5-bisphosphate (PIP2), leads to the formation of the following transmitters: diacylglycerol (DAG) and inositol triphosphate (IP3). The production of DAG induces the activity of DAG-dependent protein kinase C isoforms (PKC*δ*), which induces the MAPK/ERK pathway [161]. IP3, besides, promotes neuronal survival by repressing pro-apoptotic gene expression. This signaling cascade activates casein kinase 1 (CK1) and AKT, targeting forkhead Box G1 (FoxG1). The phosphorylation of FoxG1 by CK1 or Akt promotes neural progenitor cell differentiation into neurons in the adult hippocampus (AHN) and exerts neuroprotective effects [173,174]. Activation of the MAPK/Erk pathway conducts the transcription of CREB, Myc, and ribosomal S6 kinase (RSK), protein factors involved in synaptogenesis and neuritogenesis. The stimulation of MAPK/Erk is caused as well by the activation of Ras, which in turn is stimulated by the Src homology and containing protein (Shc)/growth factor receptor-bound protein 2 (Grb-2)/son of sevenless (SOS). Next, Erk kinase, through the RSK and MAP pathway, phosphorylates CREB and other transcription factors, leading to the modulation of the expression of genes responsible, among others, for the neurons’ survival [175]. 

Another process of brain plasticity requires the activation of the PI3K pathway generating phosphatidyl inositides, responsible for the stimulation of PKB/Akt. Akt kinase plays a crucial role in modulating the functionality of various proteins involved in the regulation of cell survival. Among others, Akt inhibits apoptosis by phosphorylation of the Bcl2-associated agonist of cell death (BAD) [161,176]. Converging the therapy on supporting neuroplastic processes seems to be a promising strategy where natural substances, especially flavonoids, can be used as enhancers of neuroplasticity on conteracting CNS diseases.

### 3.6. Neuroplasticity and Effects of Flavonoids

Other than the antioxidant and anti-inflammatory properties of flavonoids in neurodegenerative diseases, reported in previous sections of this review, recent results have revealed synaptogenesis and neurogenesis promoting properties of these compounds in CNS. Over the last decades, a *growing concern* has been devoted to the improvement of potential cognitive abilities attributed to phytochemical compounds, particularly flavonoids, and the molecular mechanisms underlying their biological action on neuronal signaling pathways. 

In an epidemiological study (PAQUID), a group of subjects (aged ≥ 65 years), free from dementia and with reliable dietary assessment for flavonoids, was examined for cognitive performance (Mini-Mental State Examination, Benton’s Visual Retention Test, “Isaacs Set Test”) raised an association between dietary flavonoid intake and better cognitive performance and evolution [177]. Flavonoids exert their effects through the modulation of neural pro-survival and apoptosis pathways, and the expression of specific genes related to neuronal differentiation, synaptic plasticity, and memory by upstreaming several transcription factors, in vitro [178,179].

Indeed, treatment with the flavonol fisetin [1 μM] in rat hippocampal slices led to an increase in the phosphorylation of approximately 2-fold ERK and 3-fold in CREB, resulting in improved memory and a long-term potentiation (LTP), a neurophysiological mechanism responsible of memory [109]. The 3,5,6,7,8,3′,4′-heptamethoxyflavone, a Citrus flavonoid, as well, exerts neuroprotective effects by inducing BDNF expression in an in vitro model of astrocytes C6 cells via the activation of cAMP/ERK/CREB signaling pathway [110]. As heptamethoxyflavone, Baicalin, a widely distributed flavone in various species of the genus Scutellaria, exercises its neuroprotective effect and improves depressive symptoms and cognitive function in models of mice with chronic unpredictable mild stress (CUMS) by regulating the ERK/CREB/BDNF signaling pathway [111]. Hesperitin, a citrus fruit flavonone, has shown a great capacity to prevent neuronal apoptosis, inhibiting apoptosis signal-regulating kinase 1 (ASK1), caspases 3 and 9 pro-apoptotic molecules by involving the activation/phosphorylation of both Akt/PKB and ERK1/2 [180]. These bibliographic data *shed light* on flavonoids’ capacities to modulate neuronal plasticity and to improve cognitive and psychomotor performance [108,181] by stimulating pathways involved in neuronal differentiation, cell survival, and inhibiting apoptosis. 

In addition, several flavonoids, such as catechins, curcumin, and myricetin, have been observed to have antidepressant-like effects in rodents and humans [112,114,182]. This suggests that flavonoids might act by increasing AHN and improving mood as well as cognitive ability. A study conducted in 2009 demonstrated the positive effects of a diet enriched in polyphenols and fatty acids (LMN diet) on neurogenesis in adult mice [183]. After 40 days on the LMN diet, an increase in several markers of AHN was found compared with mice on the control diet, with significantly more cells expressing the neuroblastic marker doublecortin (DCX) [183]. These results suggest that the LMN diet influenced neuronal differentiation. This hypothesis was confirmed by the increased co-localization of the DNA synthesis marker 5-bromo-2′-deoxyuridine (BrdU) and NeuN (postmitotic neurons marker) in the granule layer neurons of animals subjected to the LMN diet. The positive effects of the LMN diet were further confirmed by another study conducted in a mouse model of Alzheimer’s disease (Tg2576 mice) [184]. The 13-month-old mice were fed an LMN diet for 5 months and subjected during the last 2 months to various behavioral tests. It was observed that the LMN diet did not have positive effects on sensorimotor reflexes but reversed the effects of aging and especially of the Tg2576 genotype. This improvement was attributed to the 70% increase in cell proliferation in the SVZ of the brain. The neurogenic power attributed to flavonoids and the mechanism of action by which these phytochemical compounds would increase AHN were investigated in mice subjected to CUMS [185]. The animals were treated with the flavonoid quercetin, and the chronic treatment restored the weight loss of the mice caused by CUMS and alleviated CUMS-induced depression-like behaviors, such as increased sucrose consumption, ameliorate locomotor activity, and reduced immobility time. In addition, to evaluate the effects on AHN, neurogenesis markers were detected in the dentate gyrus (DG) of the hippocampus. The results showed that chronic quercetin treatment significantly increased the number of DCX (Neuronal Migration marker) and BrdU-NeuN double-positive cells (proliferating cells marker/postmitotic neurons marker), the expression levels of FoxG1, p-CREB, and BDNF in DG. These results suggest that quercetin might exert antidepressant effects through promoting AHN from the FoxG1/CREB/BDNF signaling pathway [185].

Another study conducted in 2021 shows that the flavonoid curcumin promotes hippocampal neurogenesis via the Wnt-β catenin pathway by increasing the expression of proteins involved in neurogenesis, namely, Ngn2 and NeuroD1 [113]. The mice with cerebral ischemia (CI) induced via bilateral common carotid arteries occlusion (BCCAO) were treated with curcumin, and the Morris water maze test was conducted to assess spatial learning and memory. Flavonoid treatment significantly alleviated cognitive dysfunction due to brain ischemia and promoted differentiation and maturation of new neurons in a dose-dependent manner.

These results, taken together, indicate that polyphenols, and particularly flavonoids, act by stimulating neuronal survival and proliferation, increasing AHN, and improving conditions of anxiety and depression. Thus, flavonoids could represent an important all-natural therapeutic strategy that modulates synaptic plasticity and enhances cognitive abilities in the different kinds of brain dysfunction.

### 3.7. Reduction of Neuropathological Protein Accumulation 

Neurodegenerative disorders are often associated with the accumulation of protein aggregates, which are thought to have a neurotoxic effect. Markers of neuronal degeneration in PD include Lewi bodies, cytoplasmic inclusions composed of fibrils formed from the aggregated protein α-synuclein [186]. α-synuclein is involved in numerous cytoskeletal and vesicular trafficking mechanisms, including synaptic vesicles [187]. Two identified mutations associated with early hereditary PD caused by α-synuclein loss of binding activity to vesicle [188]. In PD, the formation of 4-HNE-alpha-synuclein adduct, generated by a high level of ROS, increases the oligomerization potential, thus triggering alpha-synuclein (α-SYN) aggregation [189]. In AD studies, Aβ neurotoxicity was proved, and the implication of the amyloidogenic proteins in AD pathogenesis was demonstrated. This neurotoxicity is dependent on Aβ’s primary structure and aggregation state. Aβ1-40 and Aβ1-42 are two predominant forms comprised of 40 and 42 amino acid residues, respectively. The relative proportion of Aβ1-42 appears to be particularly responsible for AD progression, as this longer form is more predisposed to aggregation [190,191]. The aggregation of Aβ protein was associated with neuronal death. Additionally, 4-HNE is able to affect the enzymes involved in the elimination of Aβ [138]. 

Moreover, accumulation of Aβ was linked to a significant reduction in synapse number and plasticity and, consequently, to a reduced cognitive ability due to inhibition of LTP at the hippocampal level [192,193,194]. Considering the deleterious effects of neurotoxic protein aggregates in the pathogenesis of neurodegenerative diseases, different therapeutic candidates able to inhibit or reduce their accumulation and aggregation, preventing or delaying the neurodegeneration process, have been evaluated in recent years. The effects of several flavonoids on reducing protein aggregates accumulation and consequently reducing neurotoxicity have been reported in many studies [195,196]. The molecular interactions of some flavonoids with α-synuclein have been reported in an in silico study, and quercetin has shown a high affinity binding for α-synuclein [197]. This finding suggests quercetin and its analogs as potential targeted therapeutic strategies for PD. Furthermore, in a study performed on a rotenone (ROT) induced rat model of PD, apigenin significantly reduced the expression and aggregation of α-synuclein and increased the expression of dopamine D2 receptor (D2R) compared to control rats treated with ROT [105]. Regarding senile Aβ plaques in AD as well, numerous studies prove the efficacy of several flavonoids in reducing Aβ42 aggregation by inhibiting the expression of the enzyme β-secretase (BACE1) responsible for the cleavage of amyloid precursor protein (APP) and the consequent formation of Aβ42 [198,199,200,201,202]. Moreover, a study conducted on a PC12 cell line (derived from rat pheochromocytoma) showed that dihydromyricetin, a flavonoid extracted from vine tea (*Ampelopsis grossedentata*), had an inhibitory effect on Aβ fibrillation [203]. In another study conducted on Albino Wistar rats undergoing chronic mild stress, hesperidin effects on memory and learning were evaluated. Hesperidin antioxidant action and reduction of hippocampal Aβ levels and consequently the preservation of cognitive function and histological architecture of the hippocampus was proved [204].

Taken together, all these results suggest that flavonoids represent potential agents in the treatment of neurodegeneration and neurotoxicity induced by neurotoxic protein aggregates [205,206].

## 4. Zebrafish as Neurodegenerative Model in Translational Research

In the last 20 years, zebrafish have been widely used to address key questions raised in different biomedical fields: sensory systems [207,208,209,210], developmental biology [211], digestive tract [212,213,214], neurodegenerative disorders [106,215], lymphatic development [216], cutaneous wound healing [217], calcium dynamics in cardiac cells behavior [218], and for assessing new drugs [219,220]. It is a fast-expanding and extremely valuable model system whose popularity is accounted by several attributes, including breeding in captivity providing more accurate and reproducible data sets [221,222,223,224], external fertilization allowing the manipulation of embryos ex utero, optical transparency from the moment of external fertilization until organogenesis permitting the direct observation of living cells and developing tissues, using non-invasive imaging techniques [223,225], rapid development, and the availability of genomic resources. Moreover, during organogenesis, zebrafish embryos are permeable to small molecules and drugs, so to screen a drug for toxicity is fairly easy [226].

The complete zebrafish genome sequencing highlighted that 71.4% of human genes have a minimum of one zebrafish orthologue [227] and that 2601 (82%) of the total 3176 human genes bearing morbidity can be related to at least one zebrafish orthologue [228,229,230]. Indeed, zebrafish have genes orthologous to the human AD genes, including gamma secretase complex components psen1 [231], psen2 [232], ncstn (nicastrin), *aph-1* and *pen-2* [233]. Two APP homologs have been found in zebrafish, appa and appb [234]. The β secretase gene orthologues in zebrafish include bace1 [235] and bace2 [236]. The MAPT gene, encoding tau protein, has two associated orthologues in zebrafish, mapta and maptb [237], while APOE has two co-orthologues, apoea and apoeb [238]. Combined with next-generation sequencing techniques, it has revolutionized the discovery of new mutations and new candidate genes for various diseases [40], especially those involved in clinical AD and other neurodegenerative disorders [239,240,241].

Familial AD is associated with mutant *APP*, *PSEN* (Presenelin)*1* and *PSEN2* genes, E693Q mutation [242], p.Ser132Ala mutation [243], R62H mutation [244], respectively, that encode proteins for APP cleavage and Aβ generation [245]. Common genetic risk factors of sporadic AD include genetic mutant in the ε4 allele of apolipoprotein E (*APOE4*) [246], sortilin-related receptor, L 1 (*SORL1*), triggering receptor expressed on myeloid cells 2 (*TREM2*), and ATP-binding cassette transporter 7 (*ABCA7*) (p.E709fs mutation [247]) [242].

In addition, recently available tools for stable gene knockout (CRISPR/Cas9) or knockdown (morpholino) have further increased the zebrafish attractivity in the study of human disease [248], generating new transgenic models for many human diseases, namely neurodegeneration, such as familial ALS model [249], polyglutamine models [250] and tauopathy models [251,252], allowing the deduction of information on genes of interest and optimizing the discovery of preliminary notions for the improvement of the therapies currently applied. These fundamental findings and research tools invigorated the position of zebrafish as an important tool supporting mouse genetic approaches for understanding neural function in vertebrates.

APP has essential functions involving synapse formation, neural plasticity, anterograde neuronal transport, and counteracts metal-catalyzed oxidative stress [253], which explains the APP zebrafish knock-down embryos phenotype with impaired neural networks, especially in the hindbrain [254]. In zebrafish, an AD phenotype is easily induced by the insertion of the human mutant APP gene, engendering Aβ accumulation, cognitive impairments, neuronal loss, and enlarged perivascular space [255]. Hypoxia-exposed zebrafish show increased transcription of several genes, namely bace1, psen1, and psen2, linked to Aβ toxicity [235]. Additionally, bace zebrafish knockouts exhibited hypomyelination in the peripheral nervous system [236] and induced the accumulation of appa and appb proteins contributing to Aβ plaques formation [256]. Similarly, zebrafish TILLING (psen1 −/− mutants) display more histaminergic neurons associated with changes in histamine-driven behaviors [257].

Zebrafish mutants with morpholino knockdown of splicing sites in psen1 transcripts display hydrocephalic phenotype, resulting from an increased and ectopic cyclin G1 (ccng1) mRNA expression, among others [258]. The decreased PSEN1 activity through aberrant splicing of the transcript might contribute to pathological changes underlying sporadic AD [259]. The knockdown of psen2, however, induces a p53-dependent apoptotic pathway that, in turn, contributes to massive neuronal loss in zebrafish embryos [260].

Functional orthologues of the human SNCA gene encoding α-synuclein highly associated with PD pathogenesis, β-, γ1-, and γ2-synucleins have been identified in zebrafish [261]. The knockdown of β- or γ1-synucleins induces PD-like motor impairments in zebrafish [262], which are more severe when the expression of both synucleins is abolished [261]. Moreover, it was proved that the lack of both synucleins in zebrafish could induce abnormal development of the dopaminergic system, including a delay in the differentiation of dopaminergic neurons and a reduction in dopamine levels [261].

PARK is another PD-associated gene family, including PARK2 and PARK6 (PINK1), responsible for the mitochondrial motility of striatal dopaminergic neurons [263]. Morpholino knockdown of the PINK1 orthologue in zebrafish induces developmental delay and loss of neurons, principally dopaminergic ones recapitulating the human PD phenotypes better than mice [262,264,265]. PARK2 knockdown zebrafish exhibit a phenotype similar to that of PINK1 knockdown zebrafish [266]. This phenotype includes impaired mitochondrial function, loss of dopaminergic neurons in the posterior tuberculum, and increased sensitivity to the toxic effects of 1–methyl-4–phenylpyridinium (MPP+), which induces oxidative stress by disturbing the electron transport chain in mitochondrial complex I [266]. The (PARK7/DJ-1) is a Versatile protein whose protective role has been broadly validated [267]. The increased amount of overoxidized form of PARK7/DJ-1 was reported in the brain of patients with several neurodegenerative disorders, namely AD, PD, and Huntington’s disease [268]. Indeed, PARK7 knockdown in zebrafish activates apoptotic regulators, p53, and Bax proteins, which induces death of dopaminergic neurons via interaction with the mouse double minute 2 homolog protein [269].

Huntington’s disease is the most widespread monogenic neurodegenerative disorder, characterized by severe motor discoordination, cognitive deficits, and neurodegeneration [270].

IT15, the isolated homolog for the HD gene in zebrafish, shares 70% homology with its human counterpart [271]. Zebrafish with aberrant IT15 revealed neurotoxicity and abnormal HD-related protein aggregation [272,273].

Specific deletion in the 17 N-terminus amino acids (N17) area of the first HTT exon causes severe motor deficits, simulating mammalian HD-like states in zebrafish [273].

It is possible to induce a useful pharmacological model of HD by injecting quinolinic acid into zebrafish telencephalon [262].

Using zebrafish genetic models is a promising avenue of translational AD, PD, and ALS research [274]. ALS models are commonly based on mutant superoxide dismutase 1 (SOD1), TAR DNA-binding protein 43 (TDP-43), fused in sarcoma or translocated in liposarcoma (FUS), the chromosome 9 open reading frame 72 (C9orf72) gene, ubiquilin 2 (UBQLN2), matrin 3 (MATR3), and senataxin (SETX) [275]. In fact, the zebrafish SOD1 mutant shows a loss of motor neurons, muscle atrophy, and premature death [276]. In zebrafish embryos, the overexpression of mutant human SOD1 induces shortened axons and aberrant branches dose-dependent [249]. The Zebrafish ALS T70I reproduces the human ALS clinical phenotype [277]. Transgenic zebrafish expressing human TDP-43, with ALS-causative G348C mutation, recapitulate ALS phenotype by exhibiting hypolocomotion linked to spinal motor neuron axonopathy [278].

Fus is one of the RNA-binding proteins whose aggregate was associated with ALS [279]. FUS-transgenic zebrafish are characterized by a dysregulation in the cholinergic system and histone deacetylase 4, leading to the denervation and abnormal reinnervation in ALS subjects [280]. Finally, zebrafish VCP MO knockdown shows skeletal and cardiac muscle tissue degeneration via autophagy-mediated proteostasis under ALS [281].

C9orf72 is another highly promising candidate underlying ALS. A morpholino transgenic Knockdown C9orf72 zebrafish line exhibits ALS motor neuron axonopathy inducing cognitive deficits and hypolocomotion [282]. Another ALS-like phenotype transgenic line was made by injecting a DNA construct containing 89 C9orf72 hexanucleotide repeats drive in zebrafish embryos [283]. Swinnen et al. have demonstrated that the transient C9orf72 repeat RNA overexpression can be involved in the pathogenesis of C9orf72-associated ALS [284].

Moreover, mutations on the microtubule-associated protein tau (MAPT) gene have been implicated in many neurological disorders, including AD, corticobasal degeneration (CBD), progressive supranuclear palsy (PSP), Pick’s disease (PiD), chronic traumatic encephalopathy (CTE), and frontotemporal dementia with parkinsonism linked to chromosome 17 (FTDP-17) [285,286]. Studies in zebrafish have shown that expression of human tau (tau-GFP gene (htGFP) under the control of a neural-specific variant of the GATA2 promoter results within 2 days in hyperphosphorylation of tau protein and subsequent disruption of cytoskeletal structure [287]. The aggregation of hyperphosphorylated proteins is one of the characteristic neuropathological lesions of AD and other neurodegenerative disorders [288]. Recent studies have shown how the use of stable transgenic zebrafish expressing mutated human tau P301L and a fluorescent reporter (driven pan-neuronally by the HuC promoter) has allowed in vivo imaging of defective axonal growth, hyperphosphorylation of tau, and screening of novel therapeutic molecules [252]. Moreover, the AD transgenic Swedish mutant APP zebrafish with appb promoter show behavioral symptoms similar to AD and cerebral β-amyloidosis, with neuronal loss and enlarged perivascular space [255]. Another transgenic zebrafish model was made to study the development and maintenance of the blood-brain barrier due to the functional and molecular similarities with those of higher vertebrates [289]. In fact, the transgenic zebrafish line Tg (l-fabp:DBP-EGFP) expresses a vitamin D binding protein fused with an enhanced green fluorescent protein (DBP-EGFP) in the blood plasma, that could be used as an endogenous tracer for BBB breakdown allowing the monitoring of this important vascular structure in vivo [290].

The Comparative Neuroanatomy and Neurochemistry of Zebrafish NS of Relevance to Other Vertebrates NS

The main difference observed between teleosts and mammals is the neuroepithelium organization during the developmental stage that undergoes an eversion in teleost telencephalon, while mammals and non-teleost vertebrates go through an evagination [291]. Despite these, it has been shown how the zebrafish brain has functionally and morphologically overlapping zones to those of humans, such as the presence of the cerebellum, telencephalon, diencephalon, spinal cord, and enteric-autonomic nervous systems (Figure 3) [285,292,293,294]. The dorsal nucleus of the ventral telencephalic area, arising from the embryonic subpallium, is thought to be the zebrafish homologue of the mammalian striatum [295]. Same for stem niches, since the ventral nucleus of the ventral telencephalon (Vv) and the lateral zone (Dl) and/or the posterior zone (Dp) of the dorsal telencephalon of the zebrafish are considered homologous to the SVZ of the lateral ventricles and the SGZ of the DG of the hippocampus, respectively, in mammals (Figure 3).

In these areas, the levels of neurogenic potential depend on the phylogenetic derivation, which affects the distribution of progenitor cells [298,299,300]. In adult rodents, neurogenesis occurs in highly restricted spatial domains close to the forebrain ventricles, from which originate new cells destined for distinct telencephalic regions [301]. These regions are (i) the subventricular zone of the lateral ventricle leading to the generation of new interneurons reaching the olfactory and (ii) the DG in the hippocampus, where new granule neurons are produced during adulthood (Figure 3) [302,303,304,305]. In human and non-human primates, the formation of new neurons is especially evident in two regions: the SGZ and the SVZ of the telencephalon (Figure 3). Still, neurogenesis has equally been reported in several other brain regions outside the SGZ and SVZ [306], such as the basal forebrain [307], striatum [308,309], amygdala [310], SN [311], subcortical white matter [312], and at the hypothalamus [313,314,315]. Regarding synaptic transmission, teleosts show well-preserved neurotransmitter structures and systems, compared with humans, such as gamma-aminobutyric acid, glutamate, dopamine, serotonin, noradrenaline, histamine, and acetylcholine [292,316], and the endothelial blood-brain barrier similar structurally to that in higher vertebrates [289].

In contrast to mammals, zebrafish showed strong neurogenic activity supported by numerous neurogenic sites all over the brain subdivisions, including the telencephalon, the diencephalon, the mesencephalon, and the metencephalon (Figure 3). Thus, almost all sub-domains of the brain are competent for generating new neurons, while those sites are limited to only two in mammals, the hippocampus and olfactory bulb [296,297]. Both regions have consistently shown neurogenesis in all species examined thus far [317,318].

In the telencephalon, the proliferative areas are found along the ventricle in the ventral, dorsal, dorsolateral, and posterolateral domains. At the same time, in the diencephalon, they are situated in the anterior and posterior parts of the preoptic area and the anterior, mediobasal, and caudal hypothalamus. In the posterior part of the encephalon, the proliferation was reported close to the rhombencephalic ventricle. The thalamus, the pretectal periventricular region (a subdomain close to the optic tectum), the regions surrounding the habenula, and the three subdivisions of the cerebellum, including the corpus cerebelli, valvula cerebelli, and the lobus caudalis cerebelli, all harbor substantial proliferation [297,318].

The neuroepithelial cells (NECs) are the early neurogenic cells of the developing NS and transform into radial glial cells (RGCs) [319]. They are very peculiar cells with small ovoid soma whose bodies neighbor the ventricle and extend long cytoplasmic processes that reach all the path to the pial surface of the telencephalon, crossing the brain parenchyma to reach the periphery of the brain. RGCs give rise to glial progeny (oligodendrocytes and ependymal cells) but can also act as NSCs and generate almost all neurons of the brain, providing support to newly generated neurons that migrate along their radial processes [320]. They are known as true neural stem cells, able to generate new neurons through asymmetrical divisions [321,322].

In adult mammals, the RGCs maintain their neurogenic activity mainly in the SVZ of the lateral ventricles and in the SGZ of the DG of the hippocampus [323], while in the adult zebrafish telencephalon, RGCs are still able to proliferate to self-renewal and to differentiate generating new neurons [324]. Thus, the adult zebrafish appears to have kept embryonic characteristics. Indeed, the radial glial cells of adult zebrafish express similar genes and proteins to the embryonic counterparts, such as the brain lipid-binding protein (BLBP), the glial fibrillary acidic protein (GFAP), and the calcium-binding protein S100β, which is used as the standard marker for mature nongerminal astrocytes in rodents [318,325,326,327,328,329]. In adult zebrafish, up to 16 proliferating regions were detected, including those equivalents to the mammalian SVZ and SGZ, occurring from 6 months to at least 2.5 years of age [329,330,331]. Around 6000 cells are generated every 30 min in the brain of adult zebrafish, representing around 0.06% of the estimated cells of its brain [299,301,332,333]. Approximately half of the new cells escape apoptosis and survive for the rest of the fish’s life [334,335]. This is how the generation of new cells, together with the elimination of damaged cells through apoptosis, enables teleost fish rapid and efficient neuronal regeneration after brain injuries [317].

After brain injury in teleost and mammals, activated microglia, and invading leukocytes, release factors required to activate and proliferate neural stem cells, defining the injury-induced neurogenesis. An uncontrolled increase in NSCs proliferation will induce premature neurogenesis and, consequently, an early depletion of the NSC pool. However, a shallow proliferation will result in decreased neuronal compensation, which is necessary for brain homeostasis and regeneration [336].

A compared study conducted on the nuclear transcriptomic data from a zebrafish Aβ toxicity model and the datasets of two human adult brains and one fetal brain showed that approximately 95.4% of the human and zebrafish cells co-clustered [337]. Those clusters included 15 neuronal clusters (45.4% of all cells) and nine astroglial clusters (18.1% of all cells) [337]. Comparing the identified cell clusters from the zebrafish telencephalon, human entorhinal cortex, and human superior frontal gyrus separately, authors were able to determine the differentially expressed genes (DEGs) between the disease and control conditions. For instance, neuronal clusters had 801 DEGs in zebrafish and 1823 genes in the human entorhinal cortex. Among those genes, 198 were shared between both species, and 117 showed the same directionality. The AD locus, MEF2C, a protective factor against neurodegeneration, synergistically upregulated DEGs in neurons in both organisms. However, RBFOX1, an RNA-binding protein involved in Aβ clearance, was down regulated in humans, while in zebrafish neurons, it was upregulated [337]. The investigation of the molecular pathways affected in humans and zebrafish using the Kyoto Encyclopedia of Genes and Genomes (KEGG) pathway analysis of the DEGs in zebrafish neuronal clusters showed that most of the pathways are present in the human brain, including those implicated in AD [337]. However, in the astroglial clusters, authors observed more species-specific pathways [337]. Indeed, zebrafish astroglia differed from humans in the side of the neurogenic pathways [337].

Moreover, in adult zebrafish brain, the injection of human Aβ42 coupled to transportan (TR-Aβ42) form β sheets and leads to immune response and pro-inflammatory gene expression, synapses degeneration, and neurons death in zebrafish brain recording Aβ deposition-effects on mammalian brain [338]. Additionally, TR-Aβ42 administration increases progenitor proliferation leading to neurogenesis despite the prevailing toxic environment [338]. The transcriptome analysis showed that, after Aβ42 injection, the interleukin-4 (IL4) is upregulated in fish brain, suggesting the specific signaling associated with the neurodegeneration role of IL4 in zebrafish brain [338]. Moreover, Bhattarai et al. proved that the injection of interleukin-4 (IL4) through cerebroventricular promotes NSC proliferation by suppressing the tryptophan metabolism and decreasing the production of serotonin, which in turn suppresses the production of the brain-derived neurotrophic factor (BDNF) in periventricular neurons juxtaposing the NSCs [338]. In mammals, the effects of BDNF in AD are principally on neuronal survival rather than neurogenesis, while in zebrafish, BDNF directly regulates NSC plasticity [339,340]. The Aβ42 toxicity-induced neuron-glia-immune crosstalk mediated by IL4/STAT6 signaling was proved as well [341]. The same study demonstrated that the amyloid toxicity-induced interleukin-4 effects are specific to the serotonergic system [341].

These characteristics highlight the potential of zebrafish as an alternative model to better understand the constitutive and regenerative neurogenesis, as well as for the screening of new drugs that could improve the regeneration process.

## 5. Effects of Flavonoids against Neurodegenerative Disorders in Zebrafish Model

Neurodegenerative diseases comprise a wide class of disorders, such as AD, tauopathy, and α-Synucleinopathy [342], that significantly affect the quality of human life. The onset of many neurodegenerative problems depends on the interaction between genetic risk factors, environment, and aging [343,344]. The increase in the average age of the Mondial population strongly drives research in this area, combined with the limited therapeutic effectiveness of drugs currently in use due to the low bioavailability and solubility through the BBB [345]. Therefore, the scientific community is focusing on the effects of pharmacologically active molecules, medicinal herbs, and natural compounds with chemical characteristics that allow the study of new therapeutic targets. It was shown that phytochemicals, such as polyphenols and their metabolites, could pass efficiently through BBB [70,346] and also exert good effects on oxidative stress, neuroinflammation, protein aggregation, and mitochondrial dysfunction, determinants that usually increase the incidence of brain disorders [347,348,349]. In particular, the antioxidant potential of flavonoids would be a strong basis for neuron-protective activity in the brain [350,351]. Their general bioavailability and capacity to reach the brain in vivo, besides their strong binding affinity, high efficacy, less toxicity, and fewer side effects, compared to synthetic drugs [2,3], appear to play a crucial role in the expression of the neuroprotective capacity [352,353].

Together with metabolic and oxidative stress-linked diseases, neurodegenerative disorders constitute a potential application for the benefits of polyphenols [38,354,355,356,357]. Considering that most neurodegenerative diseases remain asymptomatic for nearly all the phases, the interventions initiated in advanced stages cannot repair induced damages. It has been reported that flavonoids and flavonoids rich food daily consumption, without a prior hospital diagnosis may prevent or halt the disease progression at the initial stage of the disease. Such therapies may restore neuronal function by reducing and counteracting the primary stressor. Flavonoids are crucial compounds for developing a new generation of therapeutic agents that are clinically effective in counteracting neurodegenerative disorders [358]. Studies on the effectiveness of these compounds are increasingly investigated on the zebrafish model, which has proved to be a species suitable for translational medicine research both for the genetic and functional affinity to the human NS [359]. Kim et al. showed the strong antioxidant effect of turmeric leaf extract (TLE), rich in flavonoids, due to the protective effect against H_2_O_2_-induced cell death in zebrafish embryos [360].

Similarly, in a population of zebrafish treated with H_2_O_2_ that showed cognitive impairment, the flavonoid biophenol morin significantly reduced ROS levels and increased antioxidant enzyme activity [361], improving also the behavioral activity. Two other phytochemicals, quercetin, and rutin, in addition to showing free radical scavenging activity [362,363] and anticarcinogenic effects [364], prevented scopolamine-induced memory deficits in adult zebrafish and mice [365,366], suggesting their role in the preventive strategy against Alzheimer’s progression [367]. Indeed, in another study conducted on (MPP^+^)-induced Parkinsonian-like locomotor impairment in zebrafish larvae, quercetin rescued MPP^+^-induced motor defects. This positive effect may be correlated with alleviating oxidative stress in the dopaminergic neurons by quercetin [368].

In his study published in 2019, Pan et al. created a zebrafish model with AlCl3-induced Alzheimer’s disease and tested the effects of linarine, a flavonoid glycoside from Flos chrysanthemi indici, on counteracting zebrafish AD dyskinesia. Thus, linarine’s effects on AChE inhibition were compared to donepezil (DPZ), a drug used in the AD treatment (positive control). The results reported that linarine at 50 µg/mL might significantly increase the dyskinesia recovery rate in the AD zebrafish model by 8.7% higher than DPZ at 8 µM [369]. Recently, in a zebrafish embryo model of Bisphenol A-induced neurotoxicity (BPA), the protective effect and potential mechanism of cyanidin-3-O-glucoside (C3), the most prevalent natural anthocyanin were examined [370]. The results showed that BPA-induced defects in central nervous development, downregulating neurogenesis-related genes (*Elavl3*, *Gap43*, *Zn5*, *α1-tubulin*, *Syn2a,* and *Mbp*). C3G exerts protective effects on BPA-induced neurodevelopmental toxicity through improving transcription of neurogenesis-related genes dose-dependently, enhancing antioxidative defense by reducing GSH, SOD, GPx, and CAT activity via the Nrf2/ARE pathway and reducing cell apoptosis by regulation of apoptotic genes in zebrafish larvae [370]. In addition, it has been reported that the neuroprotective mechanisms of anthocyanins involve the activation of multiple signal pathways, such as Akt, ERK1/2, CREB, BDNF, and Nrf2 signaling pathways [371].

Consistent with these results, two flavonoids, silibinin (10 µM) and naringenin (10 µM) were shown to have neuroprotective effects in zebrafish BPA-induced neurotoxicity [372]. The novel tank diving test (NTDT) and the light-dark preference test (LDPT) were performed for neurobehavioral analyses after 21 days of therapy. According to the results, co-supplementation of silibinin and naringenin significantly reduced changes in the time spent in the upper zone of the tank and the latency of entry, reduced the number of transitions to the light zone and the time spent there and thus prevented the BPA-induced change in LDPT’s scotaxis behavior [372].

Hesperidin, an abundant flavonoid of citrus fruits, interacts with several pathways involved in neurodegeneration processes, including the central CREB-BDNF pathway [373]. In a study conducted in 2021, the anticonvulsant effect of hesperidin was demonstrated in a zebrafish model of pentylentetrazole (PTZ)-induced seizures. Treatment with hesperidin significantly reduced PTZ-induced hyperactive responses and prolonged seizure latency. Indeed, the inhibition of neuronal excitation after hesperidin incubation in PTZ-exposed larvae was concomitant with decreased c-fos expression. Additionally, the treatment elevated BDNF expression and reduced interleukin-10 expression (IL-10). Those findings were supported by in silico docking analysis, which demonstrated hesperidin’s affinity for IL-10, BDNF receptor TrkB, gamma-aminobutyric acid (GABA) receptor, and N-methyl-D-aspartate (NMDA) receptor [374].

Recently the extracts of *Eucommia ulmoides* Olive (EUO), a traditional Chinese herb rich in polyphenols, were tested on zebrafish AD model, showing mitigating AD-like symptoms possibly through inhibiting excessive autophagy and the abnormal expressions of ache and slc6a3 genes [375]. In summary, polyphenols have shown promising results in zebrafish models of neuroinflammation and AD, laying the foundation for further studies on their applications. The richness of bioptives in nature, which in many cases show high permeability through the BBB, and greater neuroprotective potential, makes these compounds a resource for new therapeutic perspectives.

## 6. Conclusions

Neurodegenerative disorders, such as PD, AD, ALS, and HD, are the most worrying disorders due to the dramatic increase of affected cases, but also due to the lack of effective therapy and the presence of numerous side effects of the current therapies. Therefore, it is necessary to find an alternative therapeutic solution through translational pharmacological studies in vivo.

Zebrafish could represent the ideal experimental model to study human neurodegenerative pathologies, thanks to their high genetic and neuroanatomical homology to humans. Moreover, the high neurogenic activity maintained in the adult brain of zebrafish allows a better understanding of the regenerative and neuroprotective mechanisms exerted by flavonoids, providing potential support to develop alternative and natural therapeutic strategies against neurodegenerative diseases. Many studies have confirmed the neuropharmacological properties of flavonoids, indicating them as potential adjuvant agents in the prevention and treatment of neurodegenerative diseases. Given the abundance of these compounds in nature, research in this area should be expanded to clarify the mechanisms underlying the effects observed to date and discover possible new phytochemical compounds.

## Figures and Tables

**Figure 1 cells-12-00252-f001:**
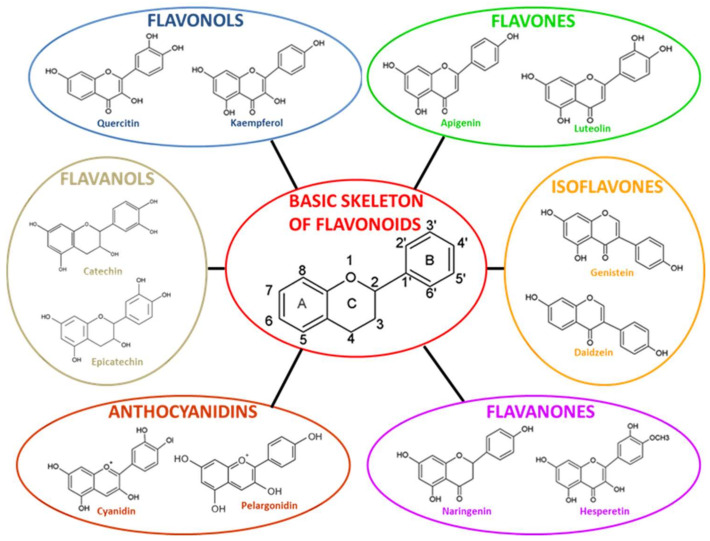
Basic skeleton structure of flavonoids and its subclasses. Adapted with permission from [38]. 2021, Giuseppe Montalbano.

**Figure 2 cells-12-00252-f002:**
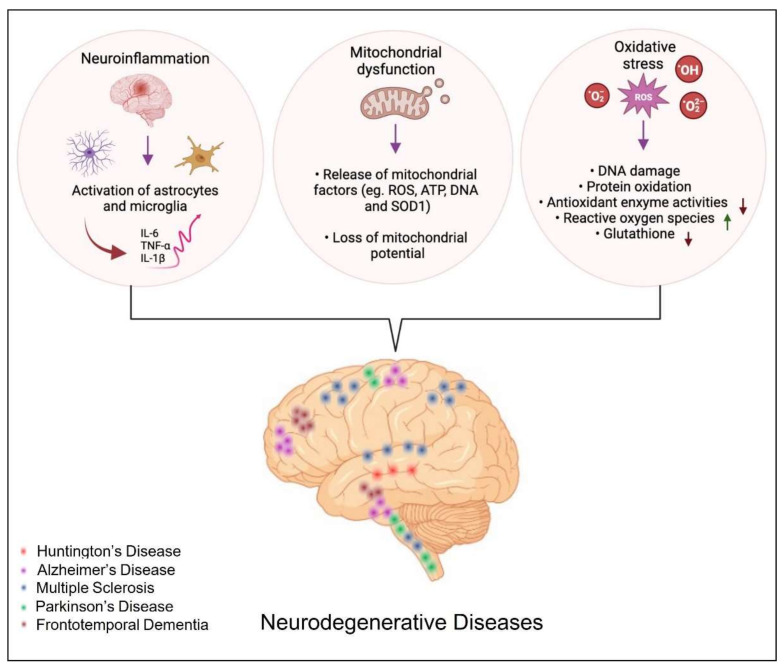
The different biological processes involved in the development of neurodegenerative diseases.

**Figure 3 cells-12-00252-f003:**
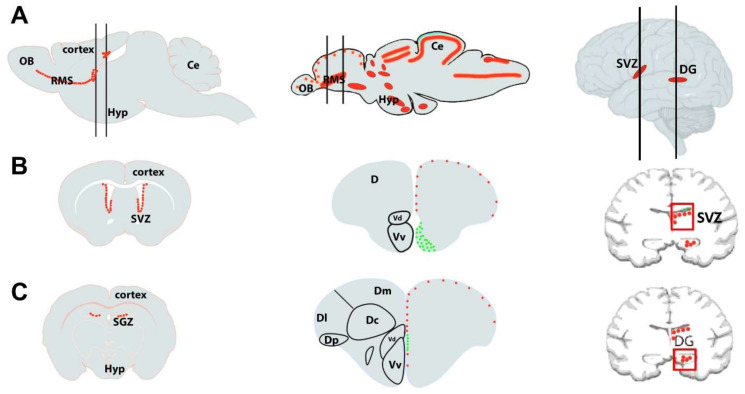
Schematic drawing of (**A**) a sagittal section of rodent (left), zebrafish (middle), and human (right) brains with the main neurogenic niches indicated in red. The mammalian brain displays only two main neurogenic niches: the subventricular zone (SVZ) of the lateral ventricles and subgranular zone (SGZ) of the dentate gyrus (DG) of the hippocampus. The black lines correspond to coronal sections. (**B**,**C**): Transversal sections through the brain, marking the major neurogenic niches of the respective species shown in A. In zebrafish, the red dots correspond to slow-cycling progenitors (mainly radial glial cells, type 2), and the green ones to fast-cycling progenitors (mainly neuroblasts, type 3). OB, olfactory bulbs; Ce, cerebellum; D, telencephalic dorsal area; Hyp, hypothalamus; RMS, rostral migratory stream; Dm, dorsomedial zone of the dorsal telencephalon; Dc, central zone of the dorsal telencephalon; Dl, lateral zone of the dorsal telencephalon; Dp, posterior zone of the dorsal telencephalon; Vd, dorsal nucleus of the ventral telencephalon; Vv, ventral nucleus of the ventral telencephalon. This figure is adapted from [296,297].

**Table 1 cells-12-00252-t001:** Bioavailability of flavonoids or flavonoid-containing foods.

Flavonoid Subclasses	C_max_	Time to Reach C_max_ (h)	The Elimination Half-Life (h)
Isoflavones	0.21–4.05	7.4–8.4	4.7–8.4
Flavonols	<0.33–7.6	<0.5–9.3	10.3–28.3
Flavanones	0.06–5.99	2–5.8	1.3–2.9
Flavanols	0.077–7.8	0.5–4.1	1–6.9
Anthocyanidins	0.011–0.041	2	-

Notes: C_max_: The maximal plasma concentration; h: hours.

**Table 2 cells-12-00252-t002:** Summary of the potential effects of some flavonoids on neurodegenerative diseases.

Flavonoid Subclass	Flavonoid	Antioxidant Potential	Anti-Inflammatory Potential	Neuroplasticity Potential	Anti- Neuropathological Protein Accumulation Potential	References
Flavonol	Quercetin	High capacity to scavenge ROS	Reduce the expression of inflammatory cytokines such as IL-6 and IL-1α	Increase of FoxG1, p-CREB and BDNF expression in DG promoting AHN	High affinity binding α-synuclein	[103,104]
Flavone	Apigenin	-	-	-	Reduced the expression and aggregation of α-synuclein	[105]
Flavanone	Hesperidin	Antioxidant action	-	-	Reduction of hippocampal Aβ levels	[70,106]
Flavonone	Hesperitin	-	-	Prevent neuronal apoptosis (inhibition of ASK1, caspases 3 and 9); Activation/phosphorylation of both Akt/PKB and ERK1/2	Affect the expression of antioxidant enzymes such as GSH and SOD by modulating the nuclear factor erythroid 2-related factor 2 (Nrf2)-ARE pathway	[107,108]
Flavonol	Fisetin	Phosphorylation of ERK and 3-fold in CREB	-	-	-	[109]
Flavones	3,5,6,7,8,3′,4′-heptamethoxyflavone	-	-	Inducing BDNF expression; Activation of cAMP/ERK/CREB signaling pathway	-	[110]
Flavone	Baicalin	-	-	Regulating the ERK/CREB/BDNF signaling pathway	-	[111]
Flavanols	Catechins	-	-	Increasing AHN	-	[112]
	Curcumin flavonoids	-	-	Increasing AHN; Promotes hippocampal neurogenesis via the Wnt-β catenin pathway; Increasing the expression of Ngn2 and NeuroD1 (involved in neurogenesis); Promoted differentiation and maturation of new neurons	-	[113]
Flavonol	Myricetin			Increasing AHN		[114]
Isoflavones	Genistein	-	Decrease the expression of Toll-Like 4 (TLR-4), inhibits microglial polarization and promoting the phenotypic switch from M1 to M2.	-	-	[115]
Flavone	Luteolin	-	Inhibit microglia activation by; Reducing the release of inflammatory mediators; Down-regulating TLR-4/TNF receptor-associated factor 6 (TRAF6)/NF-κB signaling	-	-	[116]

## Data Availability

Not applicable.

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
