# Peer review of "Zebrafish: A Model Deciphering the Impact of Flavonoids on Neurodegenerative Disorders"

_cells, 2023, doi:10.3390/cells12020252_

Round 1
Reviewer 1 Report
The manuscript by Mhalhel et al reviews the flavonoids in neurodegenerative diseases by focusing on zebrafish as a model. The topic and the contents are relevant for the journal's scope as well as it being informative for the field. However, to my opinion, there are several points that should be addressed for a more coherent and focused review. With appropriate modifications this paper is acceptable for publication.
The review is about zebrafish models and flavonoids but the reader reaches to this information at section 5, which is probably the shortest section. Mostly the chemistry of flavonoids and neurogenesis in zebrafish are discussed, which are all great, but relevant AD models of zebrafish were not included (amyloid toxicity, tau toxicity, AD related genes and their mutant versions in zebrafish etc). There is a large dataset repository for transcriptomics and proteomics in zebrafish AD research, which are not mentioned. Are there any flavonoid biology-related pathways affected in these models? How do they relate to other models mentioned in the review?
The progression from flavonoids to neurogenic outcome and inflammation and link to their antioxidant activity is plausible. The section 4 - zebrafish in neurodegeneration research contains a large section of information for neurogenesis. This is very nice but could be linked to how neurodegeneration is related to neurogenesis or regeneration with relevant literature. There is substantial information in the literature about this.
Section 5 is what the main title of the review is about but this section is the shortest. Either more information could be provided or the title could be changed.
Specific points:
Thus, several flavonoids could be used as adjuvants to pre-vent and cure neurodegenerative disorders such as Alzheimer's and Parkinson’s diseases. - This is an overstatement to my opinion. Possible involvement can be outlined but cure is a strong argument.
Over the past century, advances in biotechnology, biochemistry, and pharmacog-nosy have renewed interest in natural drugs, which have many advantages over synthetic drugs. - No references given. Justifications and examples are needed.
Zebrafish (Danio rerio) is an interesting model organism that can offer new opportunities in the research of beneficial effects of flavonoids, thanks to a combination of interesting aspects and several experimental advantages. - a rather vague sentence. What makes it interesting? Referencing needed.
NS - nervous system?
Section 3.1 - Inflammation appears to be one of the fundamental mechanisms involved in the progression of neurodegenerative diseases. - Referencing needed.
Although inflammation is not a direct trigger of neurodegenerative diseases, it is not surprising that persistent inflammation influences their progression through several mechanisms, such as initiating inflammatory responses that contribute to neuronal dysfunction and death [98] - I assume I understand the point here but inflammation has strong relationship to the pathology of the synaptic integrity, intracellular proteostasis, and other functional consequences. A more deeper account of the inflammation is needed.
Uncontrolled activation of astrocytes (neurons’ supportive cells) - astrocytes are not only supportive cells. As mentioned in large sections of the review, they form neurons as well as they control various other fundamental paradigms in the brain. Several recent reviews could be cited and incorporated here.
In overall, the statements are somehow edgy, and it is - to my opinion - better and more accommodating to include flexibility and more discussion to various sections. The literature about molecular pathways read comprehensive, yet I would suggest the authors to check the details for accuracy.
Author Response
Point-to-point reply letter
Reviewer #1
Dear Reviewer,
We would like to thank reviewer #1 for his/her time and effort in reviewing our manuscript. We greatly appreciate all the valuable comments that have helped us to improve the quality of the manuscript. We have carefully considered the comments and responded point by point to each comment. The suggested changes have been inserted into the manuscript in Microsoft Word revision mode.
Point 1:
“The review is about zebrafish models and flavonoids but the reader reaches to this information at section 5, which is probably the shortest section. Mostly the chemistry of flavonoids and neurogenesis in zebrafish are discussed, which are all great, but relevant AD models of zebrafish were not included (amyloid toxicity, tau toxicity, AD related genes and their mutant versions in zebrafish etc). There is a large dataset repository for transcriptomics and proteomics in zebrafish AD research, which are not mentioned. Are there any flavonoid biology-related pathways affected in these models? How do they relate to other models mentioned in the review?’’
Response 1:
Thank you for pointing out the incompleteness of this section. Section 5 has been developed (L853-L861, L873-L912) and different AD models of zebrafish (L629-L635 and L645-L718), the related genes (L620-L625) as well as the different mutants have been reported.
Point 2:
“The progression from flavonoids to neurogenic outcome and inflammation and link to their antioxidant activity is plausible. The section 4-zebrafish in neurodegeneration research contains a large section of information for neurogenesis. This is very nice but could be linked to how neurodegeneration is related to neurogenesis or regeneration with relevant literature. There is substantial information in the literature about this.”
Response 2:
In the new version of the manuscript, the link between neurodegeneration and neurogenesis is added with the relevant citations (L824-L829).
Point 3:
“Section 5 is what the main title of the review is about, but this section is the shortest. Either more information could be provided, or the title could be changed.”
Response 3:
Section 5 has been developed in the revised manuscript (L853-L861, L873-L912).
Specific points:
Point 1:
“Thus, several flavonoids could be used as adjuvants to pre-vent and cure neurodegenerative disorders such as Alzheimer's and Parkinson’s diseases. - This is an overstatement to my opinion. Possible involvement can be outlined but cure is a strong argument.”
Response 1:
The effects of flavonoid on the neurodegenerative diseases’ pathology are reported in (L19-L21). Nevertheless, the word ‘cure’ was replaced by ‘counteract’ (L22).
Point 2:
“Over the past century, advances in biotechnology, biochemistry, and pharmacog-nosy have renewed interest in natural drugs, which have many advantages over synthetic drugs. - No references given. Justifications and examples are needed.”
Response 2:
References and the justification of the reported affirmation have been added (L35-L36).
Point 3:
“Zebrafish (Danio rerio) is an interesting model organism that can offer new opportunities in the research of beneficial effects of flavonoids, thanks to a combination of interesting aspects and several experimental advantages. - a rather vague sentence. What makes it interesting? Referencing needed.”
Response 3:
The reported sentence was used to introduce the zebrafish model. The different attributes making zebrafish a good model in the research of beneficial effects of flavonoids in neurodegenerative diseases are explained extensively in (L604-L832).
Point 4:
“NS - nervous system?”
Response 4:
It is true that NS is an abbreviation for the nervous system, we have modified the description in our revised manuscript (L91).
Point 5:
“Section 3.1 - Inflammation appears to be one of the fundamental mechanisms involved in the progression of neurodegenerative diseases. - Referencing needed.”
Response 5:
Reference has been added to the above-mentioned sentence (L285).
Point 6:
“Although inflammation is not a direct trigger of neurodegenerative diseases, it is not surprising that persistent inflammation influences their progression through several mechanisms, such as initiating inflammatory responses that contribute to neuronal dysfunction and death [98] - I assume I understand the point here, but inflammation has strong relationship to the pathology of the synaptic integrity, intracellular proteostasis, and other functional consequences. A more deeper account of the inflammation is needed.”
Response 6:
The relationship between inflammation and both synaptic integrity and intracellular proteostasis was added both in (L288-L293 and L330-L340) and in Section 3.7 (L560-L569).
Point 7:
“Uncontrolled activation of astrocytes (neurons’ supportive cells) - astrocytes are not only supportive cells. As mentioned in large sections of the review, they form neurons as well as they control various other fundamental paradigms in the brain. Several recent reviews could be cited and incorporated here.”
Response 7:
The reported sentence was revised, and references were added in L292-L295 in the revised manuscript.
Point 8:
“In overall, the statements are somehow edgy, and it is - to my opinion - better and more accommodating to include flexibility and more discussion to various sections. The literature about molecular pathways read comprehensive, yet I would suggest the authors to check the details for accuracy.”
Response 8:
Thank you for your valuable comments, we have edited the manuscript and checked the different parts.
Reviewer 2 Report
The review „Zebrafish: A Model deciphering the Impact of Flavonoids on Neurodegenerative disorders” by Mhalhel, Sicari, Pansera et al., is a very detailed and comprehensive summery of the current knowledge of Flavonoids’ effect on neurodegeneration and neurogenesis. The review is well structured and provides a detailed summary on Flavonoids, their structure, subclasses and their known biological effects. The special focus of this review is on neurodegeneration and several mechanisms known to influence neurodegeneration including neuroinflammation, ROS production, neurogenesis and protein aggregation have been carefully mentioned and discussed. A nice table summarizing the current status of published effects of Flavonoids on neurodegenerative diseases is provided.
The review is well written and will be valuable resource to the scientific community doing research in ths field.
I fully support the publication of this report.
I only have minor comments that should be addressed:
Line 53: better citation for AD: Kheradmand et al., Biomed Pharmacother. 2080.
Line 270: Neurodegenerative diseases affect nerve cells and the nervous system (NS) impacting motor, sensory and/or cognitive functions.
Neurodegenerative diseases also can affect non- neuronal cells , please rephrase accordingly!
Line 444: …Akt promotes neural progenitor cells differentiation into neurons in …
Correct: …..Akt promotes neural progenitor cell differentiation into neurons in…
Line 472: …"Isaacs Set Test) ….
Correct: …"Isaacs Set Test“) …
Line 540: there is an alpha missing in front of -synuclein
Line 544: Ab42, an oligomeric forms of Aβ known for its potent neurotoxicity,
This statement is incorrect! Aß42 is NOT an oligomeric form of Aß.
Maybe the authors ment to write Aß42, and oligomeric forms of Aß
Please clarify and bear in mind that also other Aß species than Aß42 can be toxic!
Line 560: …In …. Should not be …in….
Legend of Figure 3 is missing explanations of many abbreviations!
Line 719 and 720: stress prevention and effect on nutrition here more. We now have the first positive effects on cognition with an Aß targeting antibody! Mention and point out the even greater potential with combination therapy and the easy availability of a Flavonoid therapy even without a specific diagnosis which would be a prerequisite for an antibody therapy!
In summary: this is a very well written and well researched comprehensive review that should be accepted!
Author Response
Point-to-point reply letter
Reviewer #2
Dear Reviewer,
We would like to thank you for taking the time and effort to review our manuscript. We sincerely appreciate the valuable comments and suggestions that have helped us to improve the quality of this article. Our responses to your comments are described point by point below. Changes made in response to your suggestions have been incorporated into the manuscript and are highlighted.
Point 1:
“Line 53: better citation for AD: Kheradmand et al., Biomed Pharmacother. 2080.”
Response 1:
The reference has been added.
Point 2:
“Line 270: Neurodegenerative diseases affect nerve cells and the nervous system (NS) impacting motor, sensory and/or cognitive functions.
Neurodegenerative diseases also can affect non- neuronal cells, please rephrase accordingly!”
Response 2:
The mentioned sentence was rephrased, and the appropriate reference has been added (L272-L273).
Point 3:
“Line 444: …Akt promotes neural progenitor cells differentiation into neurons in …
Correct: …. Akt promotes neural progenitor cell differentiation into neurons in…”
Response 3:
The reported spelling error has been corrected (L461).
Point 4:
“Line 472: …"Isaacs Set Test) …
Correct: …"Isaacs Set Test") …”
Response 4
The missing quotation mark has been added (L490).
Point 5:
“Line 540: there is an alpha missing in front of -synuclein”
Response 5:
The missing alpha has been added in our revised manuscript (L577).
Point 6:
“Line 544: Ab42, an oligomeric form of Aβ known for its potent neurotoxicity, this statement is incorrect! Aß42 is NOT an oligomeric form of Aß. Maybe the authors ment to write Aß42, and oligomeric forms of Aß. Please clarify and bear in mind that also other Aß species than Aß42 can be toxic!”
Response 6:
This reported sentence has been rephrased (L622-L628).
Point 7:
“Line 560: …In …. Should not be …in….”
Response 7:
We guess the reviewer was trying to correct the typo “In” in our manuscript. It should be lowercase “in” rather than “In”, so we modified in the revised manuscript (L586).
Point 8:
“Legend of Figure 3 is missing explanations of many abbreviations!”
Response 8:
The authors are thankful for the reviewer’s comment. The missing abbreviations in Figure 3 were added (L764-L767).
Point 9:
“Line 719 and 720: stress prevention and effect on nutrition here more. We now have the first positive effects on cognition with an Aß targeting antibody! Mention and point out the even greater potential with combination therapy and the easy availability of a Flavonoid therapy even without a specific diagnosis which would be a prerequisite for an antibody therapy!”
Response 9:
The potential of flavonoid as an effective therapy agent for counteracting neurodegenerative disorders was added in the revised manuscript (L853-L861).
Round 2
Reviewer 1 Report
Majority of my comments were addressed.
The Alzheimer's disease-related studies in zebrafish are not completely accounted for. Several significant studies that determined molecular mechanisms in zebrafish brain with amyloid as well as comparisons to human brains and investigation of human disease genes in zebrafish with clinical samples are not cited. This is a deficiency in this review. Authors can give a final effort if they wish.
Author Response
Dear Reviewer,
We would like to thank reviewer for his/her time and effort in reviewing our manuscript. We greatly appreciate all the valuable feedback that have helped us to improve the quality of the manuscript. We have carefully considered the insightful comment and updated our manuscript properly.
Point 1:
“The Alzheimer's disease-related studies in zebrafish are not completely accounted for. Several significant studies that determined molecular mechanisms in zebrafish brain with amyloid as well as comparisons to human brains and investigation of human disease genes in zebrafish with clinical samples are not cited. This is a deficiency in this review. Authors can give a final effort if they wish.’’
Response 1:
Answer 1:
We modified the text according to the reviewer's suggestion and added 2 references describing molecular mechanisms in zebrafish brain with amyloid and comparison with human brain in section 4.1 (L826-861).